# Fixed-dose ivermectin for Mass Drug Administration: Is it time to leave the dose pole behind? Insights from an Individual Participant Data Meta-Analysis

Adriana Echazu[1,2], Daniela Bonanno[1], Pedro Emanuel Fleitas[3], Julie Jacobson[4], Gabriela Matamoros[5], Charles Mwandawiro[6], Wendemagegn Enbiale[7], Áuria de Jesus[8], Alan Brooks[9] Alejandro Javier Krolewiecki[1,2]*

1 Fundación Mundo Sano, Madrid, Spain, 2 Instituto de Investigaciones de Enfermedades Tropicales (IIET), Universidad Nacional de Salta, Orán, Salta, Argentina, 3 Instituto de Salud Global de Barcelona (ISGlobal), Barcelona, Spain, 4 Bridges to Development, Vashon, Washington, United States of America, 5 Instituto de Investigaciones en Microbiología (IIM), Facultad de Ciencias, Universidad Nacional Autónoma de Honduras, Tegucigalpa, Honduras, 6 Kenya Medical Research Institute (KEMRI), Nairobi, Kenya, 7 Dermatovenerology Department, College of Medicine and Health Science, Bahir Dar University, Bahir Dar, Ethiopia, 8 Centro de Investigação em Saúde de Manhiça (CISM), Manhiça, Maputo, Mozambique, 9 Bridges to Development, Geneva, Switzerland

* alekrol@mundosano.org

## Abstract

### Background

Ivermectin (IVM) is widely used in mass drug administration (MDA) programs for the control of neglected tropical diseases (NTDs). Current regimens rely on weight- or height-based dosing, which lead to operative challenges. This study evaluates an age-based fixed-dose regimen for IVM.

### Methodology

This is an individual participant data (IPD) meta-analysis including anthropometric data from over 700,000 individuals, across 53 NTD-endemic countries. Fixed-dose regimens were developed based on weight distribution by age. The proportion of individuals achieving the target range dose (200–400 µg/kg) was assessed and compared to traditional dosing regimens.

### Principal Findings

Fixed-doses of 3 mg for pre-school children (PSAC), 9 mg for school-aged children (SAC), and 18 mg for women of reproductive age (WRA) resulted in a higher proportion of participants receiving the target dose compared to weight- and height-based regimens (79.9% vs. 32.7% and 37.3%, respectively, p < 0.001). Underdosed individuals were fewer with fixed-dose (8.7%) compared to weight-based (32.6%) and

**Data availability statement:** All relevant data are available from the Figshare repository at the following DOI: 10.6084/m9.figshare.28734971. Available at: https://figshare.com/articles/dataset/_b_Fixed-Dose_Ivermectin_for_Mass_Drug_Administration_Is_it_time_to_leave_the_dose_pole_behind_Insights_from_an_Individual_Participant_Data_Meta-Analysis_b_/28734971?file=53591264.

**Funding:** This study was conducted as part of the STOP 2030 project, funded by the Global Health EDCTP3 Joint Undertaking and its members, supported by the European Union's Horizon Europe research and innovation programme and was co-funded by the Federal Department of Economic Affairs, Education and Research (EAER) and the State Secretariat for Education, Research and Innovation (SERI) of the Swiss Confederation. The grant agreement number, which is shared by both the main funder and the co-funder, is Nº 101103089. As per funding requirements, the supporting organizations should be acknowledged in the following order: (1) Horizon Europe Health (EDCTP3) and (2) Swiss Confederation. The funders had no role in the design of the study, data collection or analysis, interpretation of results, or the decision to submit the manuscript for publication.

**Competing interests:** I have read the journal's policy and the authors of this manuscript have the following competing interests: AJK is the Principal Investigator of the EDCTP-funded STOP2030 project, which is coordinated by Liconsa Laboratories. In this role, he holds an ad-hoc consultancy with Liconsa Labs. All other authors have declared that no competing interests exist.

height-based (46.3%) regimens. Although doses above the target range increased slightly, most remained within 600 μg/kg.

## Conclusions

An age-based fixed-dose regimen for IVM could improve treatment coverage and simplify MDA activities. Simplified logistics could lead to cost savings in drug distribution and administration, improving the overall efficiency of MDA programs. These findings support the inclusion of currently excluded PSAC in IVM-based MDA interventions. More broadly, this paper provides evidence for considering the potential policy and programmatic implications of fixed-dose IVM. This Individual Participant Data Meta-analysis (IPD-MA) is registered in PROSPERO (CRD42024521610).

## Author summary

Ivermectin is an essential drug with proven safety and effectiveness against several parasitic infections. It plays a key role in Mass Drug Administration (MDA) programs targeting prevalent Neglected Tropical Diseases. Currently, ivermectin dosing is based on weight or height, which can be difficult to measure in the field during MDA campaigns and adds complexity and workload for health workers. These methods also carry a risk of underdosing.

In this study, we analyzed data from more than 700,000 participants across 53 countries to explore whether a fixed-dose approach could simplify MDA implementation while maintaining doses within the therapeutic range. We found that fixed-dose regimens provide more accurate treatment for a larger proportion of individuals, reduce the likelihood of underdosing, and only occasionally result in doses above the recommended levels, typically by small margins and in a limited proportion of participants. This simplified approach could ease treatment delivery in community settings and improve coverage and operational efficiency. Our findings provide practical evidence to inform policy discussions on how to streamline and strengthen ivermectin-based MDA programs.

## Introduction

The control of several neglected tropical diseases (NTD) is based on the provision of single-dose anthelmintic drugs through mass drug administration (MDA) campaigns [1,2]. Ivermectin (IVM) is an essential medicine widely used in MDA activities [3]. It is the drug of choice against onchocerciasis, scabies and strongyloidiasis, and in combination with other anthelmintics, against lymphatic filariasis and trichuriasis [4–7].

IVM is typically administered using a weight-based dosing regimen, with the currently recommended dose being 200 μgr/kg of body weight. In MDA programs, the World Health Organization (WHO) recommends using height as a proxy for weight,

implemented through a dose pole to facilitate large-scale delivery [1,8]. Under this approach, children shorter than 90 cm or weighing less than 15 kg are excluded from treatment. Early clinical trials evaluating the safety and efficacy of ivermectin for onchocerciasis demonstrated that single doses as high as 800 µg/kg were as safe as lower doses although some adverse events like edematous swelling and subjective ocular troubles were more frequent at doses of 800 µg/kg in subjects with onchocerciasis [9,10]. Notably, while these studies support the safety of higher doses, a clear upper safety limit for ivermectin in humans has not yet been formally established; therefore doses nearing 800 µg/kg should be approached with caution, especially when used for onchocerciasis.

Current IVM dosing strategies pose operational challenges in large-scale MDA programs. Several studies have shown that height or weight-based dosing often leads to sub-therapeutic treatment in certain populations [11,12]. Additionally, children appear to have an accelerated clearance of IVM, potentially requiring higher doses [13]. Despite its favourable safety profile, data on IVM use in children under 5 years of age remain limited. Nevertheless, a systematic review including 1088 children under 15 kg of body-weight that received IVM for a variety of indications suggests a safety profile comparable to that observed in older children [14]. This evidence coupled with accumulating information on the safety of IVM at doses several times higher than currently recommended, supports the exploration of alternative fixed-dose regimens [15–17]. The European Medicines Agency's Positive Opinion supporting age-based fixed-dosing of 9 and 18 mg of IVM in combination with albendazole aligns with this broader safety margin for IVM [18].

With over 500 million people receiving IVM annually through MDAs alone, the potential public health benefits of a fixed-dose approach are substantial [19,20]. This analysis aims to identify an alternative age-based dosing regimen using anthropometric data to evaluate fixed-dose regimens and compare drug exposure across current and exploratory strategies.

## Methods

### Ethics Statement

This retrospective observational study did not require ethical approval, as it involved the analysis of de-identified data from previously conducted studies. All data were fully anonymized, ensuring that individual participants could not be identified and that confidentiality was maintained. The study relied on simulated ivermectin doses applied to hypothetical scenarios; no actual medication was administered, eliminating risks to any participants. Additionally, the study was conducted in accordance with established ethical standards, including the guidelines of the Council for International Organizations of Medical Sciences (CIOMS) in collaboration with the WHO, and the principles outlined in the Declaration of Helsinki [21].

### Study design and registration

This is an Individual Participant Data Meta-analysis (IPD MA). The protocol has been registered at the International prospective register of systematic reviews PROPSERO: (PROSPERO 2024 CRD42024521610): https://www.crd.york.ac.uk/prospero/display_record.php?ID=CRD42024521610.

### Eligibility criteria

Studies were eligible if they provided individual-level data on participants' anthropometric measurements and geographic location, regardless of the original study objectives or reported outcomes.

Study-Level Inclusion Criteria:

1. Study design: Observational studies, public health surveys, and clinical trials.

2. Geographic scope: Countries or sub-national districts where preventive chemotherapy for STH is recommended according to WHO guidelines [1,22].

3. Time frame: Studies conducted between January 1, 2010, and December 31, 2024.

4. Data availability: Studies including anthropometric data. Participants with missing or incomplete IPD for any mandatory variable (country, age, sex, and weight) were excluded. Individuals with missing height data were retained, as height was considered non-essential.

5. Timing of data collection: Only baseline data was included.

Study-Level Exclusion Criteria:

1. Studies providing only aggregated data.

2. Studies enrolling severely ill subjects, such as those focused on tuberculosis or severe malaria.

Individual-Level Inclusion Criteria:

1. Age Groups: pre-school age children (PSAC): 2–4 years (24–59 months); school age children (SAC): 5–15 years; woman of reproductive age (WRA): 15–49 years.

2. Sex: Male and female (PSAC and SAC); only female adults (WRA). Data for adult males were unavailable [1].

**IPD collection process and data integrity**

IPD were obtained from two main sources: (1) studies on STH interventions conducted in endemic regions and (2) datasets from data repositories, accessed in compliance with their specific protocols.

The datasets underwent a five-step process: selection, standardization, compilation, cleaning, and consistency assessment. First, only variables of interest were retained, excluding others from the original studies. Next, standardization ensured consistency across datasets: age was recorded in months (children) or years (adults), weight in kilograms (one decimal), and height in centimeters (one decimal). Each site was coded by country.

This study assumed population homogeneity, justifying a one-stage meta-analysis, and during the compilation step, IPD from different studies were merged into three datasets: PSAC, SAC, and WRA. Cleaning followed, removing subjects with missing data. Finally, a consistency assessment was conducted using WHO Anthro (version 3.2.2) and WHO Anthro Plus (version 1.0.4) to detect potential measurement errors. Data were flagged as inconsistent if z-scores exceeded pre-defined thresholds:

- PSAC: WHZ < -5 or > 5; WAZ < -6 or > 5; HAZ < -6 or > 6; BAZ < -5 or > 5.

- SAC & WRA: WAZ < -6 or > 5; HAZ < -6 or > 6; BAZ < -5 or > 5.

These cut-offs, set by the software, were applied to identify extreme values likely resulting from measurement errors while accounting for diverse nutritional scenarios [23].

**Risk of bias assessment**

The IPD compiled for this study were collected regardless of the design or outcomes of the original studies. Only baseline raw anthropometric IPD were included, rendering risks of bias related to outcome reporting or randomization processes irrelevant. Data completeness and consistency across studies were assessed, and participants with missing or inconsistent IPD were excluded according to pre-defined criteria applied systematically across all datasets. Since weight and height measurements were not conducted by our team, potential measurement errors, inaccuracies, or data entry mistakes could not be prevented. To minimize measurement bias, data flagged as inconsistent by the software were excluded. However, this approach may introduce selection bias if exclusions were not proportionally distributed across

sites or other characteristics. The potential impact of these exclusions on the representativeness of the sample was considered in the analysis.

## Statistical analysis

The IPD were compiled into a database using Microsoft Excel (Microsoft, Redmond, WA). Data analysis was conducted with R version 3.1.1 (The R Foundation for Statistical Computing, GNU General Public License). Heterogeneity analysis was performed with the 'metafor' package [24]. Graphics were generated using the 'ggprism' extension of the 'ggplot2' package [25]. Map was created using QGIS version 3.43.10 (QGIS Development Team, released under the GNU General Public License, Version 2).

We applied a random effects model to address potential residual heterogeneity across the datasets included in the analysis and to examine the variation in effects across different subgroups. We assessed statistical heterogeneity using the $I^2$ statistic and $\tau^2$, which allowed us to quantify the degree of variability.

Nutritional status of children was assessed using stunting proportions with 95% confidence intervals (CI) by country. Stunting was defined as a Height-for-Age Z-score (HAZ) below -2 SD from the international reference median. HAZ was selected as it applies to children from birth to 19 years. HAZ is also the best indicator of chronic malnutrition, often linked to STH infections. Malnutrition severity was classified by stunting prevalence using WHO thresholds: < 20% (mild), 20–29% (moderate), 30–39% (high), and >40% (very high) [26].

The IPD were analyzed through a sequential process for the exploration of IVM dosing regimens as follows:

### 1) Selection of age-based fixed-dose

The distribution of weight was analyzed to assess its dispersion. Each participant's weight was multiplied by 200 to determine the exact IVM dose required to achieve the recommended 200 µg/kg. The median IVM dose was calculated for each year of age. Based on this median, the fixed-dose for each target group was determined by selecting the number of standard 3 mg tablets closest to the median dose required for the upper age limit of that group.

### 2) Dose Calculation

The dose of IVM (in µg/kg) for each participant was calculated using three methods: i) Weight-based; ii) Height-based (dose-pole); iii) Fixed-dose as determined in Step 1. The weight and height- based doses used were those recommended by WHO [1,13].

| Ivermectin dosing method | Not currently recommended | 3 mg | 6 mg | 9 mg | 12 mg | 15 mg |
|---|---|---|---|---|---|---|
| Weight-based | < 15 Kg | 15 – 24 kg | 25 – 35 kg | 36 – 50 kg | 51 – 65 kg | 66 – 79 kg |
| Height-based | < 90 cm | 90 – 119 cm | 120 – 139 cm | 140 – 159 cm | >159 cm | |

### 3) Dose Categorization

Participants were categorized into six groups based on the dose received under each regimen:

- Not currently recommended: For weight-based regimen, body weight <15 kg; for height-based regimen, height <90 cm.

- Underdosed: < 200 µg/kg.

- Target Range: 200–400 µg/kg.

- Above the Target Range (Level 1): 401–600 µg/kg.

- Above the Target Range (Level 2): 601–800 µg/kg.

- Above the Target Range (Level 3): > 800 µg/kg.

4) Comparison Between Regimens

The proportion of participants in each dosing category (not currently recommended, undertreated, correctly treated, or above the target range) was calculated for each treatment regimen. To assess differences in proportions across regimens, pairwise comparisons with Chi-square test ($\alpha = 0.05$) was performed. Additionally, median doses across the three regimens were compared using the Related Samples Friedman Test, stratified by group (PSAC, SAC, WRA). If significant differences were found, Wilcoxon signed-rank tests with Bonferroni correction were applied for post-hoc pairwise comparisons.

### Outcomes

The primary outcome of this study was: proportion of participants receiving target range dose of IVM (200–400 µg/kg) with a fixed-dose regimen based on age. A secondary outcome was the comparison of the performance of the fixed-dose regimen with height or weight-based dosing methods. The measure of effect for this outcome were proportion and median doses differences between dosing regimens.

Additional secondary outcomes were prevalence of undernutrition in children globally and by country in the study population and identification of a threshold age at which ivermectin administration may be contraindicated.

## Results

### IPD database sources

IPD were gathered from various original study datasets through the following sources: 1) Three studies focused on soil-transmitted helminths (STH) interventions conducted in Africa and Latin America [16,17,27]; 2) One study accessed via the Harvard Dataverse data repository [28]; 3) One dataset downloaded from the Digital Commons@Becker data repository [11]; 4) Datasets from 48 countries were requested to the Demographic and Health Surveys (DHS) Program [29]; and 5) Datasets from 57 studies were requested from the Infectious Diseases Data Observatory (IDDO) [30–88]. The participant flow diagram is shown in Fig 1, and additional details on the included studies are provided in the supporting information (S1 Table). During data checking, we identified issues in some individual records, such as missing values for age or weight, implausible height or weight measurements, or formatting errors in age registration. We addressed these problems by applying predefined exclusion criteria to remove affected participants from the dataset. These criteria were applied systematically across the full dataset and were not specific to any study, country, or subgroup. Importantly, the distribution of excluded participants was similar across countries, and no significant geographical selection bias was introduced. No entire study was excluded from the analysis; only individual participants were removed based on the criteria described. After the selection process, a total of 741,700 participants from 53 countries were included. The countries where preventive chemotherapy is recommended by WHO and those with available data to be represented in the study are highlighted in Fig 2.

### Characteristics of the study population

The study population included 398,376 WRA and 343,324 children, of whom 173,205 (50.4%; 95% CI: 50.3–50.6) were female. Among children, 317,401 (92.4%; 95% CI: 92.3–92.5) were PSAC, while 25,923 (7.55%; 95% CI: 7.4–7.6) were SAC. In accordance with WHO definitions, the WRA category begins at 15 years of age [89]. In this analysis, 15-year-old participants were included in both the SAC and WRA groups: in the SAC group to reflect data availability for both sexes, and in the WRA group, which included only females. Presenting this age in both categories highlights that 15 years could serve as a practical cut-off age for transitioning between fixed-dose regimens in MDA programs. Heterogeneity analysis

PLOS Neglected Tropical Diseases

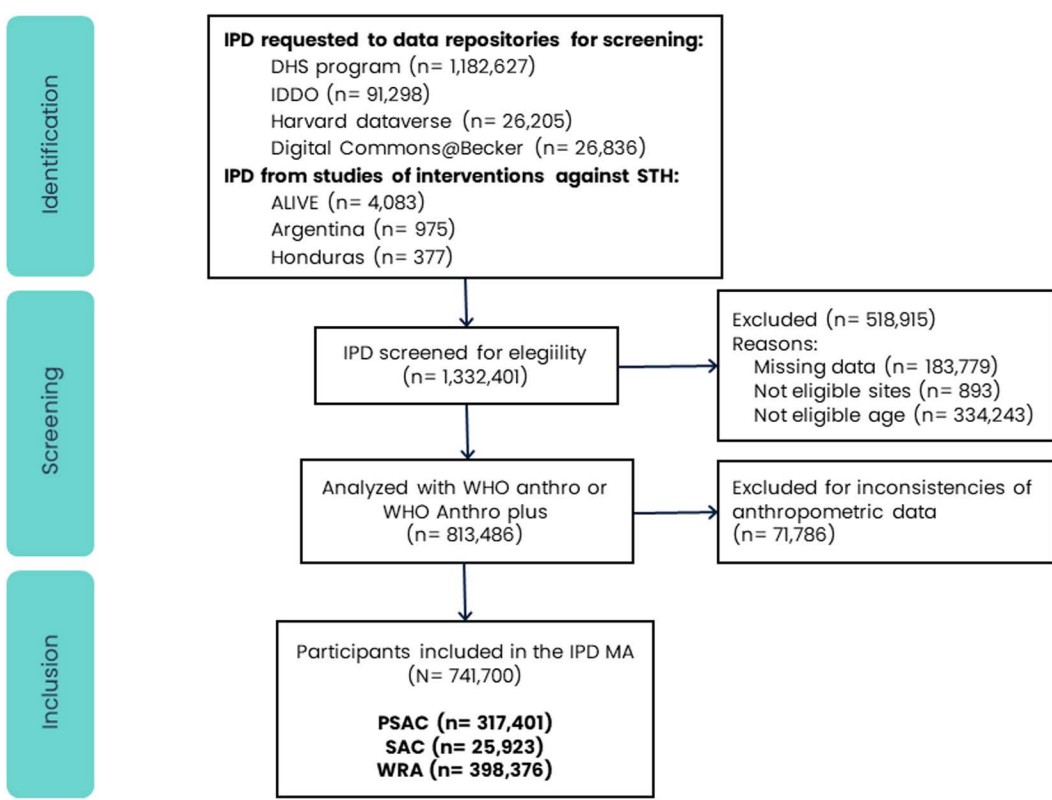

**Fig 1. Flow diagram of participants of the Individual participant data meta-analysis of ivermectin fixed-dose feasibility.**

showed I² values of 17%, 18%, and 15% for PSAC, SAC, and WRA, respectively, indicating minimal heterogeneity. τ² values (0.005, 0.01, and 0.01) further confirmed low variability, suggesting that despite structural differences, heterogeneity across datasets was minimal.

Nutritional status assessment revealed that 120,718 children (35.2%; 95% CI: 35–35.3) were stunted (HAZ<-2). Consequently, the severity of malnutrition, as indicated by stunting prevalence, was classified as high across all countries (S2 Table) [26].

### Ivermectin dose exploration

**Fixed-dose selection.** The distribution of weight by age was analyzed, revealing a homogeneous sample. Among PSAC, the mean weight was 12.93 (SD: 2.45) kg, with a median of 12.70 kg (IQR: 11.2–14.4). In SAC, the mean was 27.25 (SD: 11.08) kg, and the median 24.80 kg (IQR: 19–33). For WRA, the mean was 54.99 (SD: 11.96) kg, and the median 52.90 kg (IQR: 46.8–60.9).

The weight-based calculation identified a median IVM dose of 2.54 mg (IQR: 2.24–2.88) for PSAC; 4.96 mg (IQR: 3.8–6.6) for SAC; and 10.58 mg (IQR: 9.36–12.18) for WRA, required to achieve the dose of 200 µg/kg. Percentiles of the calculated IVM dose by year of age for children and by group for WRA are presented in Table 1 (also in S1 Fig).

Based on these findings, fixed-dose regimens of 3 mg for PSAC and 9 mg for SAC were chosen to approximate the median required dose for the upper age limit of each group of children (median value for 4 yo for PSAC, median value for 15 yo for SAC). For WRA, an 18 mg fixed-dose was selected to approximate the 95th percentile for this group, minimizing

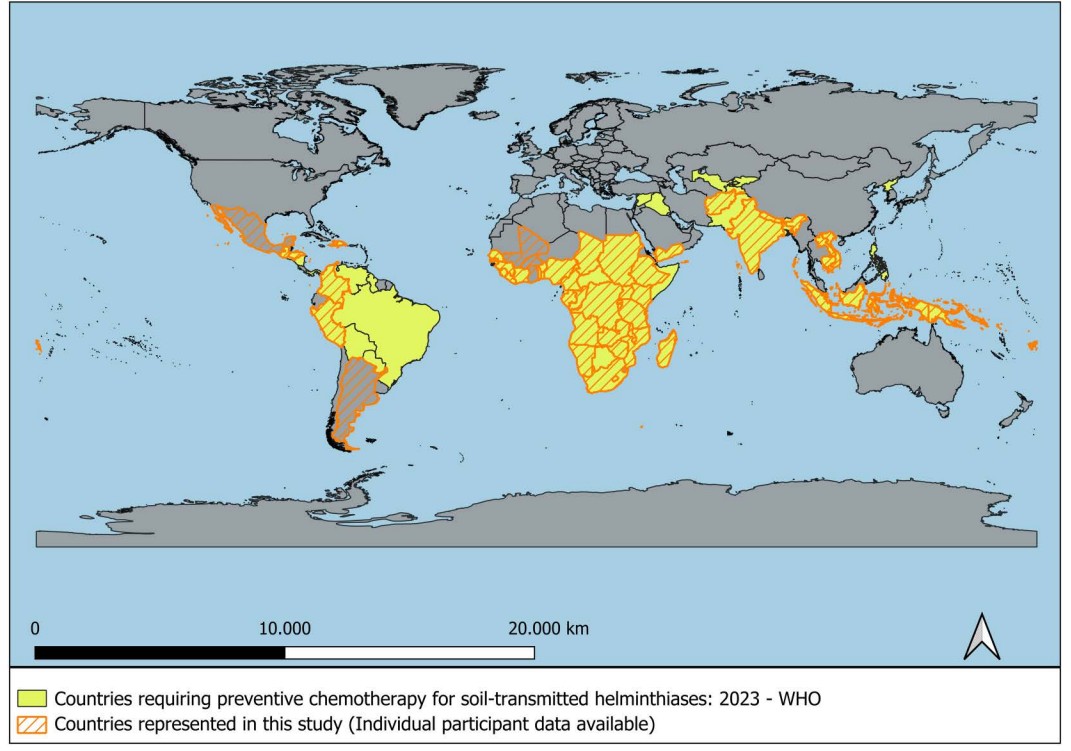

**Fig 2. Map showing the countries included in the Individual Participant Data Meta-analysis (N = 741,700).** Note: Map created in QGIS. Base map data from Natural Earth (public domain). Available at: https://www.naturalearthdata.com. Map of countries requiring preventive chemotherapy for soil-transmitted helminthiases in 2023 adapted from WHO (https://apps.who.int/neglected_diseases/ntddata/sth/sth.html).

**Table 1. Percentiles of calculated amount of ivermectin required for reaching the recommended dose (200 µg/kg) by age (N = 741,700).**

| Target Group | AGE (years) | 5th Percentile (mg) | 10th Percentile (mg) | 25th Percentile (mg) | 50th Percentile (mg) | 75th Percentile (mg) | 90th Percentile (mg) | 95th Percentile (mg) |
|---|---|---|---|---|---|---|---|---|
| PSAC (n = 317,401) | 2 | 1.72 | 1.82 | 2.02 | 2.22 | 2.46 | 2.70 | 2.88 |
| | 3 | 2.00 | 2.10 | 2.32 | 2.56 | 2.84 | 3.10 | 3.30 |
| | 4 | 2.24 | 2.40 | 2.60 | 2.88 | 3.18 | 3.50 | 3.72 |
| SAC (n = 25,923) | 5 | 2.40 | 2.80 | 3.00 | 3.20 | 3.60 | 4.00 | 4.40 |
| | 6 | 2.80 | 2.80 | 3.20 | 3.60 | 4.00 | 4.63 | 5.00 |
| | 7 | 3.00 | 3.06 | 3.43 | 4.00 | 4.52 | 5.05 | 5.60 |
| | 8 | 3.20 | 3.40 | 3.90 | 4.40 | 5.00 | 5.61 | 6.20 |
| | 9 | 3.42 | 3.80 | 4.20 | 4.80 | 5.60 | 6.26 | 7.00 |
| | 10 | 3.80 | 4.00 | 4.60 | 5.20 | 6.00 | 7.00 | 7.80 |
| | 11 | 4.20 | 4.60 | 5.03 | 6.00 | 6.87 | 8.00 | 9.00 |
| | 12 | 4.40 | 4.80 | 5.20 | 6.12 | 7.40 | 8.80 | 10.00 |
| | 13 | 5.00 | 5.41 | 6.20 | 7.28 | 8.60 | 10.20 | 11.39 |
| | 14 | 5.53 | 6.00 | 7.00 | 8.00 | 9.40 | 10.94 | 12.30 |
| | 15 | 6.20 | 6.80 | 7.60 | 8.80 | 10.00 | 11.60 | 12.94 |
| WRA (n = 398,376) | 15–45 | 7.94 | 8.42 | 9.36 | 10.58 | 12.18 | 14.06 | 15.46 |

Note: Dose calculated for each participant with the formula: weight (kg) * 0.2 mg.

the risk of underdosing (S3 Table). The median dose of IVM with the selected fixed-dose regimen by age group was 236 µg/kg (IQR: 208–267) for PSAC; 363 µg/kg (IQR: 272–473) for SAC and; 340 µg/kg (IQR: 295–384) for WRA. The highest variability observed was in the SAC group. Fig 3A displays the distribution of IVM dose by age group using the fixed-dose regimen with median and IQR.

### Comparison between dosing regimens

Considering the entire study population (PSAC, SAC, and WRA), the fixed-dose regimen resulted in the highest proportion of participants receiving the target range dose (79.9%), compared to the weight-based (32.7%) and height-based regimens (37.3%), (p < 0.001). Conversely, the fixed-dose regimen had the lowest proportion of participants classified as underdosed (8.7%), with this difference also being statistically significant compared to the other regimens (p < 0.001) (Table 2). The proportion of participants classified as receiving doses above the recommended range was significantly higher with the fixed-dose regimen (11.27%) compared to the weight-based and height-based dosing methods, which had proportions of 0% and 0.003%, respectively. Among participants who received doses above the target range, most fell

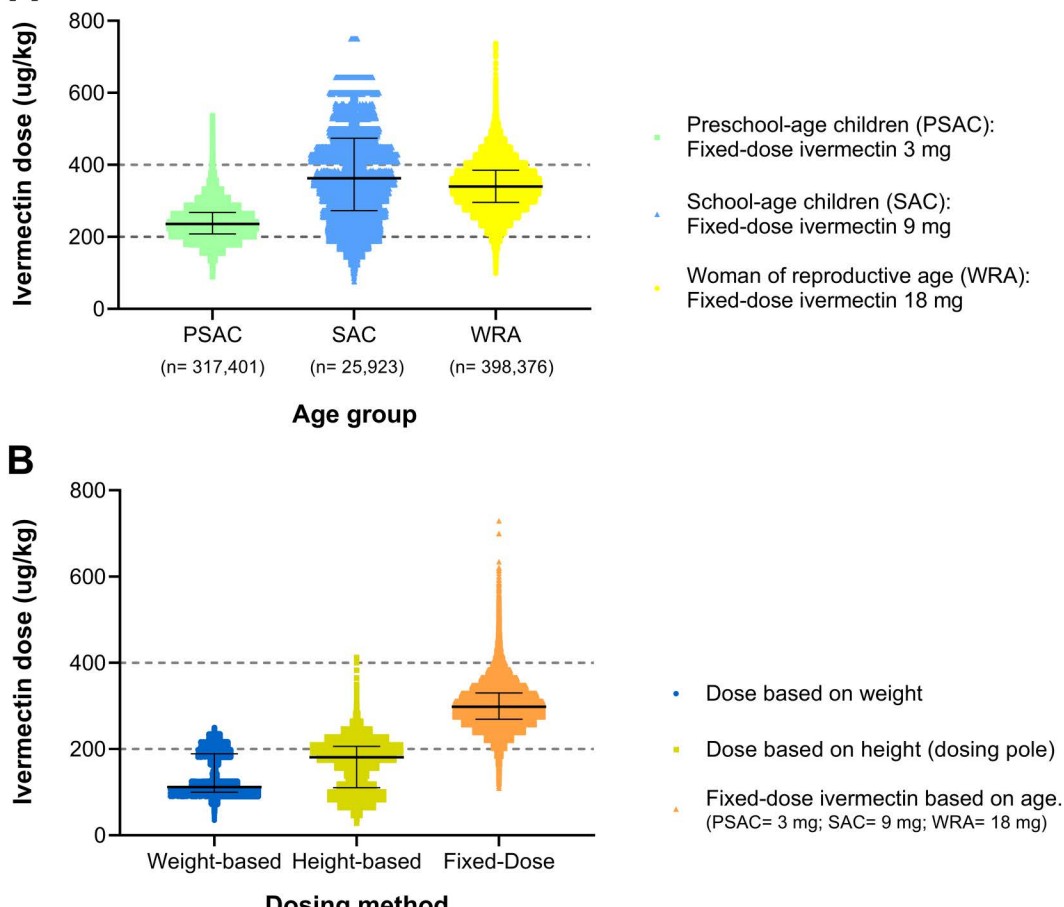

**Fig 3. Ivermectin dose (µg/kg) distribution across dosing regimens.** Panel A: Ivermectin dose using the fixed-dose regimen by age group (PSAC, SAC, and WRA); Panel B: Distribution of ivermectin doses across the study population, comparing weight-based, height-based, and fixed-dose methods (N = 741,700).

Table 2. Proportion of participants within each dosing category by dosing regimen (N = 741.700).

| Dose Category | Weight-Based (WB) % (95% CI) | Height-Based (HB) % (95% CI) | Fixed-dose (FD) % (95% CI) | p-value p1: WB vs HB p2: WB vs FD p3: HB vs FD |
|---|---|---|---|---|
| Not currently recommended* | 34.7 (34.6 – 34.8) | 16.4 (16.3 – 16.5) | NA | p1: < 0.001 |
| Underdosed (< 200 µgr/kg) | 32.6 (32.5 – 32.7) | 46.3 (46.2 – 46.5) | 8.7 (8.6 – 8.8) | p1; p2; p3: < 0.001 |
| Target Range Dose (200 – 400 µgr/kg) | 32.7 (32.5 – 32.8) | 37.3 (37.2 – 37.4) | 79.9 (79.8 – 80) | p1; p2; p3: < .001 |
| Above the Target Range Dose Level 1 (401 – 600 µgr/kg) | 0 | 0.003 (0.002 – 0.004) | 11.1 (11 – 11.2) | p1; p2; p3: < 0.001 |
| Above the Target Range Dose Level 2 (601 – 800 µgr/kg) | 0 | 0 | 0.15 (0.14 – 0.16) | p1: 1 - p2; p3: < 0.001 |
| Above the Target Range Dose Level 3 (< 800 µgr/kg) | 0 | 0 | 0.02 (0.016 – 0.022) | p1: 1 - p2; p3: < 0.001 |

Notes: * Not currently recommended: For weight-based regimen, body weight <15 kg; for height-based regimen, height <90 cm. NA: No exclusion criteria were applied for the age-based fixed-dose regimen.

within level 1 (400–600 µg/kg). However, 0.02% exceeded the highest threshold (>800 µg/kg, level 3), with the maximum recorded dose reaching 1125 µg/kg. Fig 3B illustrates the distribution of IVM doses across the study population, presenting the median and interquartile ranges for each dosing approach, with most participants receiving doses below the target range of 200–400 µg/kg with both weight-based and height-based regimens. Fig 4A, 4B and 4C display the proportion of participants by dose category and by dosing regimen for each age group.

Differences in the proportion of participants receiving the target IVM dose with the fixed-dose regimen were analyzed across countries. No significant variation was found between countries with a high prevalence of malnutrition and those without. Moreover, countries with severe stunting had fewer underdosed participants with fixed-dose compared to other countries.

In the comparison of median IVM doses, the fixed-dose regimen (median dose: 298; IQR: 269–330 µgr/kg) consistently provided a higher dose compared to the weight-based (median dose: 112; IQR: 99.5–189) µgr/kg) and height-based (median dose: 181; IQR: 110–206 µgr/kg) regimens. Additionally, current regimens failed to achieve the intended 200 µg/kg dose. In contrast, the fixed-dose regimen consistently maintained doses within the target range. In PSAC, the fixed-dose regimen provided a higher dose than both the height-based (Z = -321.021, p < 0.001) and weight-based regimens (Z = -438.539, p < 0.001). In SAC, the fixed-dose regimen yielded a higher dose than the height-based (Z = -63.915, p < 0.001) and weight-based regimens (Z = -126.721, p < 0.001). In WRA, the fixed-dose regimen again provided a higher dose compared to both the height-based (Z = -245.987, p < 0.001) and weight-based regimens (Z = -546.611, p < 0.001). Since all comparisons yielded consistently low p-values (p < 0.001), Bonferroni correction was not required.

### Sex differences

In the PSAC group, the median weight was 13 kg (IQR: 3.2 kg) for boys and 12.5 kg (IQR: 3.2 kg) for girls. A significant difference in weight distribution between sexes was observed, as confirmed by the Mann-Whitney U test (p < 0.001). In the SAC group, boys had a median weight of 24 kg (IQR: 13 kg), while girls had a median weight of 25 kg (IQR: 14 kg). A statistically significant difference was also identified in this group (Mann-Whitney U test, p = 0.046). In the PSAC group, 79.6% of boys and 83.8% of girls received the recommended dose of IVM with the fixed dose of 3 mg. Additionally, 0.2% of boys and 0.4% of girls received doses above the recommended range (200–400 µg/kg), with these differences being statistically significant (Fisher's exact test, p < 0.001). No child in the PSAC group, regardless of sex, received a dose above 600 µg/kg.

In the SAC group, 50.9% of boys and 51.5% of girls achieved the recommended dose using the fixed dose of 9 mg; this difference was not statistically significant (Fisher's exact test, p = 0.154). However, when analysing doses above the

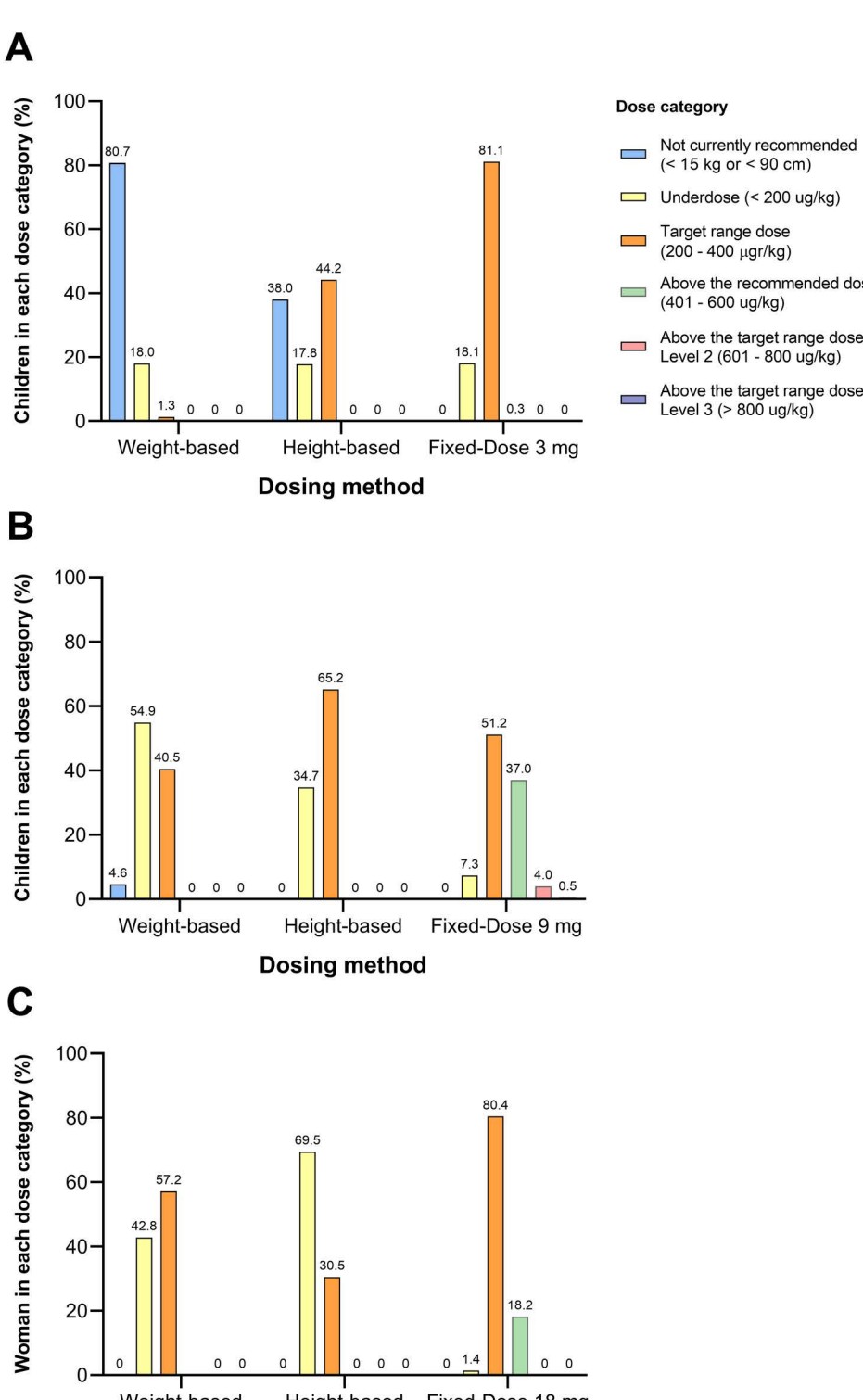

**Fig 4. Proportion of participants per dose category and by dosing method for each age group.** Panel A: Pre-school age children (PSAC; n = 317,401); Panel B: School age children (SAC; n = 25,923); Panel C: Woman of reproductive age (WRA; n = 398,376).

recommended (400 µg/kg), 42% of boys and 40% of girls exceeded this threshold with a statistically significant difference (p = 0.039). Within this group, only 0.5% of boys and 0.6% of girls received doses above 800 µg/kg, and this difference was not statistically significant (p = 0.232). Sex-based differences are presented in S2 and S3 Figs.

### PSAC subgroup analysis

Among PSAC, 80.7% (95% CI: 80.6–80.9) of children were excluded from IVM use based on the weight-based regimen, as their weight was below 15 kg. In contrast, 37.9% (95% CI: 37.8–38.2) were excluded based on the height-based approach. This difference was statistically significant (p < 0.001). Fig 5A illustrates the correlation between weight and height among PSAC participants. Fig 5B further highlights some lack of concordance between these two criteria, as the intersection of a 15 kg weight and 90 cm height aligns with the 95th percentile.

If a fixed 3 mg dose were administered to all PSAC without applying any anthropometric criteria for exclusion, the median dose delivered would be 236 µg/kg (IQR: 446 µg/kg). Under this approach, 81.1% (95% CI: 81.5–81.8) of children would receive a dose within the recommended range. Less than 0.3% (95% CI: 0.2–0.3) would exceed 400 µg/kg, and none would surpass 600 µg/kg.

Regarding the threshold age for ivermectin administration, children typically reach 15 kg between 4 and 5 years of age (median weight: 14.4 kg at 4 yo and 16 kg at 5 yo). However, our findings suggest that children aged 2 years and older could receive a 3 mg dose without exceeding the 600 µg/kg upper limit, irrespective of current contraindication criteria. Fig 5C depicts the median dose achieved by age in PSAC under a fixed 3 mg dosing regimen.

## Discussion

This IPD meta-analysis, which included 741,700 participants from 53 NTD endemic countries, provides a comprehensive assessment of fixed-dose ivermectin regimens. By modelling drug exposure under different dosing methods and calculating the dose that each participant would receive, our analysis demonstrates that an alternative age-based fixed-dose regimen achieves therapeutic dosing in a higher proportion of individuals compared to weight- and height-based regimens. Furthermore, we found that a simplified dosing approach reduces systematic underdosing without a great risk of exceeding established safety thresholds. This alternative approach, aligned with existing public health classification of at-risk groups (PSAC, SAC and WRA), could simplify drug administration logistics while contributing to dose optimization [1].

Compared to weight- or height-based approaches, an age-based fixed-dose regimen reduces wide-spread underdosing and, significantly increases the proportion of adequately treated individuals, with small risk of doses over 800 µg/kg. Our findings align with those of Goss (2019) regarding IVM underdosing in adult populations and further expand the evidence of this issue in school-aged children (SAC). This concern was also highlighted by Buonfrate (2023), who found that fewer than 50% of SAC participants achieved the optimal dose with the current dosing approaches [11,90].

Available data indicate that IVM doses of 600 µg/kg, and possibly 800 µg/kg, are safe and suggest that the target dose range for IVM is due for reconsideration [15–17]. This supports the feasibility of implementing age-based fixed-dose regimens in PSAC, SAC, and adults, ensuring doses remain within the therapeutic index. In adults, trials have demonstrated the safety of single doses up to 2,000 µg/kg in healthy volunteers and daily doses of 1,200 µg/kg for five consecutive days in COVID-19 patients [91,92].

While these findings support the feasibility of higher and age-based fixed-dose regimens, safety remains a key concern, particularly regarding the potential risks associated with exceeding recommended dose thresholds. Although ivermectin does not normally cross the blood–brain barrier, there is a theoretical risk of increased permeability in individuals with severe malnutrition, potentially elevating the risk of central nervous system toxicity. Despite the safety profile of ivermectin is well established, with millions of doses administered worldwide, serious neurological events have been reported, mostly in individuals with high-intensity Loa loa infections or during treatment for onchocerciasis [93]. Additionally, rare

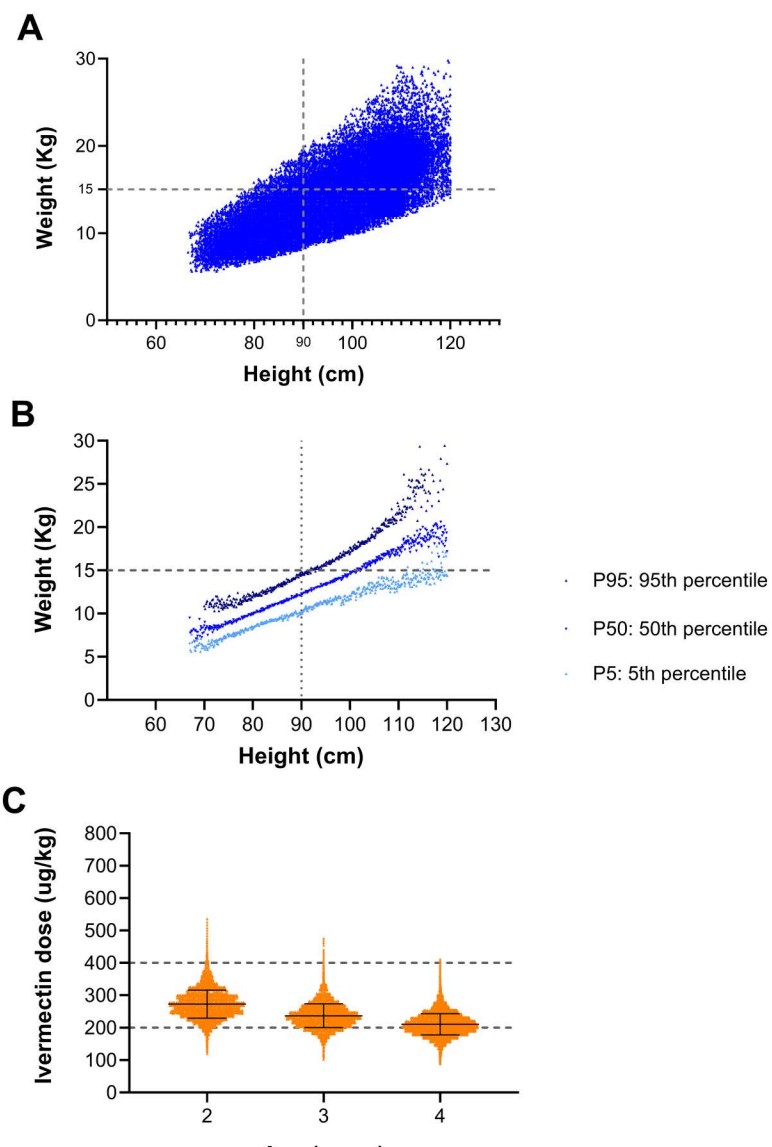

**Fig 5. (n = 317,401): Panel A: Correlation between height and weight of pre-school age children (PSAC); Panel B: weight for height percentiles of PSAC participants of the study; Panel C: Distribution of ivermectin dose (µg/kg) by year of age in PSAC using the fixed-dose regimen.** Notes for Fig 5, Panels A and B: Gray dashed lines indicate the current lower limits of the weight-based and height-based recommendations for ivermectin use. Panel B: Percentiles 5, 50, and 95 represent the distribution of weight by height among the study population of pre-school aged children, corresponding to the lower, median, and upper percentiles respectively.

genetic mutations affecting drug transport mechanisms can predispose individuals to severe adverse effects and could become more relevant in large-scale MDA campaigns [94]. These considerations highlight the importance of including robust pharmacovigilance systems (both active and passive) as an essential component of any intervention using fixed-dose strategies based on age, despite the logistical and financial challenges they may entail.

Regarding potential sex-based differences in dosing, our stratified analysis indicated that while some differences were statistically significant, they are unlikely to be clinically relevant. No clear pattern emerged suggesting a disproportionate

risk of misdosing between males and females under age-based fixed-dose regimens. This finding aligns with current ivermectin dosing methods, which do not differentiate between sexes.

PSAC are systematically excluded from ivermectin treatment based on age, weight, or height criteria. However, these exclusion thresholds are inconsistently applied and not clearly aligned with the actual risk of overdosing. For example, age-based criteria often exclude children under five, even when their weight exceeds 15 kg [7]. Similarly, weight- and height-based regimens do not align with standard growth curves, resulting in poor concordance and the unnecessary exclusion of eligible children. This is especially concerning given the high burden of diseases like soil-transmitted helminthiases and scabies in PSAC, and their increased vulnerability to related morbidity. Our findings suggest that a fixed 3 mg dose would achieve adequate drug exposure in most PSAC, while addressing concerns about potential overdosing [13,95]. The observed drug exposure by age aligns with the systematic review by Jittamala et al., which found a comparable safety profile in children weighing below and above 15 kg [14]. Revisiting current exclusion criteria could prevent unnecessary barriers to treatment and support more equitable access of young children to ivermectin-based interventions.

A key strength of this study is the large number of participants, all from countries endemic for STH and other NTDs. The participants represent "real-world" recipients of MDA interventions, characterized by a high prevalence of malnutrition. As a result, evaluating fixed-dose IVM in this population enhances confidence in the low risk of excessive dosing. It is also relevant the homogeneity of the results across geographic regions.

This study has several limitations. SAC were underrepresented compared to PSAC and WRA, which may affect the generalizability of findings for this group. Despite SAC receiving the highest volume of anthelmintic drugs globally through school-based MDA programs targeting STH, anthropometric data remain scarce. Additionally, determining a fixed dose for SAC was challenging due to weight variation, as growth in this age range is steady. The selected dose prioritized minimizing underdosing.

The study relied on secondary data and the sources of IPD was diverse, which may have introduced inconsistencies or missing values, though heterogeneity analysis showed overall variability was low. Countries were assumed to be similar to be analyzed as a single population, but site-specific factors, such as India's unique nutritional and regulatory context, warrant further investigation of those sites, beyond the scope of this study.

Findings may also lack applicability to adult males, as no IPD were available for this group. Nonetheless, given that NTDs cause substantial morbidity in men and impact productivity in endemic regions, adult males would also likely benefit from IVM treatment [96,97]. The fixed dose for WRA was set at the upper weight range, ensuring adequate treatment and may also provide appropriate dosing to adult males.

The timing of these findings add urgency to the growing data suggesting that the target dose range for IVM should be revisited. Further research will continue to contribute to our collective understanding, these results offer a strong foundation for considering a shift in practice to a fixed-dose regimen. WHO and national NTD programs may identify specific implementation research needs to assess the cost-effectiveness, and practical application of this approach in endemic settings, alongside its acceptability among healthcare workers and communities. Additionally, pharmaceutical development of paediatric formulations ensuring palatability and bioequivalence for PSAC remains an important research gap [98,99]. The upper safety limit for ivermectin dosing has not yet been clearly established. Further studies, including analyses of data from its widespread use during the COVID-19 pandemic, could help define this limit more precisely [91]. However, any call for additional studies should be weighed against the risk of prolonging the widespread use of sub-therapeutic IVM dosing, which could delay meaningful improvements in treatment outcomes.

In conclusion, the findings of this study provide robust evidence to inform policy discussions on IVM dosing, supporting the feasibility and benefits of transitioning from weight- and height-based IVM dosing to an age-based fixed-dose regimen. They offer critical insights into drug exposure among PSAC, a group currently excluded from MDA interventions. A fixed-dose strategy would reduce the substantial proportion of underdosed individuals without increasing the risk of toxicity [18].

Taken together, these findings present a compelling case as policy-makers at international, regional, and national levels consider updating treatment guidelines. The potential public health benefits -greater efficiency, broader coverage, and improved community engagement-underscore the importance of translating this evidence into action.

## Supporting information

**S1 Table. Summary of extracted data and sources for included studies.**
(PDF)

**S2 Table. Nutritional assessment by country.**
(PDF)

**S3 Table. Ivermectin dosing regimens.** Currently recommended wight-based and height-based and alternative age-based fixed-dose.
(PDF)

**S4 Table. List of contributors to IDDO dataset.**
(PDF)

**S1 Fig. Distribution of IVM dose by age.**
(TIF)

**S2 Fig. Ivermectin dose (µg/kg) by sex in pre-school-age children (PSAC) receiving a fixed 3 mg dose.** Median and interquartile range (n = 317,401).
(TIF)

**S3 Fig. Ivermectin dose (µg/kg) by sex in school-age children (SAC) receiving a fixed 9 mg dose.** Median and inter-quartile range (n = 25,923).
(TIF)

**S1 PRISMA Checklist. PRISMA-IPD Checklist.**
(PDF)

## Acknowledgments

We extend our acknowledgment and gratitude to all authors and institutions that contributed their valuable data.

This research includes data obtained through a request to the Infectious Diseases Data Observatory (IDDO) (https://www.iddo.org/data-sharing/accessing-data). IDDO had no role in the production of this research. It also includes data provided by the COVID-19 Data Platform, which is hosted and led by IDDO and the International Severe Acute Respiratory and emerging Infections Consortium (ISARIC), both of which had no role in the production of this research.

Permission to reuse the data was requested by IDDO from the original contributor, or from the Data Access Committee if the responsibility had been delegated by the contributor. This permission was a prerequisite for data sharing. A full list of the organisations and authors that contributed the data in the dataset requested for the present study to the IDDO Data Platform is provided in S4 Table.

We would like to especially acknowledge Shanti Rochester (shanti.rochester@iddo.org), Data Governance Officer at the Infectious Diseases Data Observatory (IDDO), Big Data Institute Building, University of Oxford, for her kind assistance during the data sharing process.

We also gratefully acknowledge Professor Bob Taylor(Bob@tropmedres.ac) and Dr. James Watson (jwatson@oucru.org) for their valuable advice on data collection strategies.

## Author contributions

**Conceptualization:** Adriana Echazú, Pedro Emanuel Fleitas, Julie Jacobson, Alan Brooks, Alejandro J Krolewiecki.

**Data curation:** Adriana Echazú, Daniela Bonanno, Pedro Emanuel Fleitas, Gabriela Matamoros.

**Formal analysis:** Adriana Echazú, Daniela Bonanno.

**Funding acquisition:** Alejandro J Krolewiecki.

**Investigation:** Adriana Echazú, Daniela Bonanno, Julie Jacobson, Gabriela Matamoros, Mwandawiro Charles, Wendemagegn Enbiale, Auria De Jesus, Alejandro J Krolewiecki.

**Methodology:** Adriana Echazú, Daniela Bonanno, Pedro Emanuel Fleitas, Alan Brooks, Alejandro J Krolewiecki.

**Project administration:** Alejandro J Krolewiecki.

**Resources:** Alejandro J Krolewiecki.

**Software:** Adriana Echazú, Daniela Bonanno.

**Supervision:** Adriana Echazú, Pedro Emanuel Fleitas, Julie Jacobson, Alan Brooks, Alejandro J Krolewiecki.

**Validation:** Adriana Echazú, Julie Jacobson, Mwandawiro Charles, Wendemagegn Enbiale, Auria De Jesus, Alan Brooks, Alejandro J Krolewiecki.

**Visualization:** Adriana Echazú, Daniela Bonanno, Pedro Emanuel Fleitas, Gabriela Matamoros, Mwandawiro Charles, Wendemagegn Enbiale, Auria De Jesus, Alan Brooks, Alejandro J Krolewiecki.

**Writing – original draft:** Adriana Echazú, Daniela Bonanno, Alan Brooks, Alejandro J Krolewiecki.

**Writing – review & editing:** Adriana Echazú, Daniela Bonanno, Pedro Emanuel Fleitas, Julie Jacobson, Gabriela Matamoros, Mwandawiro Charles, Wendemagegn Enbiale, Auria De Jesus, Alan Brooks, Alejandro J Krolewiecki.

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
