## [Decision Letter · Decision Letter 0]

3 Jul 2025

PNTD-D-25-00581

Fixed-Dose Ivermectin for Mass Drug Administration: Is it time to leave the dose pole behind? Insights from an Individual Participant Data Meta-Analysis.

Dear Dr. Krolewiecki,

Thank you for submitting your manuscript to PLOS Neglected Tropical Diseases. After careful consideration, we feel that it has merit but does not fully meet PLOS Neglected Tropical Diseases's publication criteria as it currently stands. Therefore, we invite you to submit a revised version of the manuscript that addresses the points raised during the review process.

Please submit your revised manuscript within 60 days Sep 01 2025 11:59PM. If you will need more time than this to complete your revisions, please reply to this message or contact the journal office at plosntds@plos.org. Please include the following items when submitting your revised manuscript:

We look forward to receiving your revised manuscript.

Kind regards,

Michael Marks

Academic Editor

Francesca Tamarozzi

Section Editor

Shaden Kamhawi

co-Editor-in-Chief

Paul Brindley

co-Editor-in-Chief

**Journal Requirements:**

At this stage, the following Authors/Authors require contributions: Adriana Echazú, Daniela Bonanno, Pedro Emanuel Fleitas, Julie Jacobson, Gabriela Matamoros, Mwandawiro Charles, Wendemagegn Enbiale, Auria De Jesus, Alan Brooks, and Alejandro Javier Krolewiecki. Please ensure that the full contributions of each author are acknowledged in the "Add/Edit/Remove Authors" section of our submission form.

Potential Copyright Issues:

i) Figure 2. Please (a) provide a direct link to the base layer of the map (i.e., the country or region border shape) and ensure this is also included in the figure legend; and (b) provide a link to the terms of use / license information for the base layer image or shapefile. We cannot publish proprietary or copyrighted maps (e.g. Google Maps, Mapquest) and the terms of use for your map base layer must be compatible with our CC BY 4.0 license.

1) State the initials, alongside each funding source, of each author to receive each grant. For example: "This work was supported by the National Institutes of Health (####### to AM; ###### to CJ) and the National Science Foundation (###### to AM).".

6) Your current Financial Disclosure states, "Yes ↳ Please add funding details. This research is part of the STOP2030 project, supported by the Global Health EDCTP3 Joint Undertaking and its members, and has received funding from the European Union’s Horizon Europe research and innovation programme under grant agreement Nº 101103089. STOP2030 is also funded by the Federal Department of Economic Affairs, Education and Research (EAER) and the State Secretariat for Education, Research and Innovation (SERI) of the Swiss Confederation.The funders had no role in study design, data collection/analysis, interpretation or the decision to submit the manuscript for publication. ↳ Please select the country of your main research funder (please select carefully as in some cases this is used in fee calculation). SPAIN - ES".

However, your funding information on the submission form indicates only "HORIZON EUROPE Health". 

Please indicate by return email the full and correct funding information for your study and confirm the order in which funding contributions should appear. Please be sure to indicate whether the funders played any role in the study design, data collection and analysis, decision to publish, or preparation of the manuscript.

7) As required by our policy on Data Availability, please ensure your manuscript or supplementary information includes the following:

8) Please send a completed 'Competing Interests' statement, including any COIs declared by your co-authors. If you have no competing interests to declare, please state "The authors have declared that no competing interests exist". Otherwise please declare all competing interests beginning with the statement "I have read the journal's policy and the authors of this manuscript have the following competing interests:"

**Reviewers' Comments:**

Reviewer's Responses to Questions

**Key Review Criteria Required for Acceptance?**

**Methods**

-Are the objectives of the study clearly articulated with a clear testable hypothesis stated?

-Is the study design appropriate to address the stated objectives?

-Is the population clearly described and appropriate for the hypothesis being tested?

-Is the sample size sufficient to ensure adequate power to address the hypothesis being tested?

-Were correct statistical analysis used to support conclusions?

-Are there concerns about ethical or regulatory requirements being met?

Reviewer #1: The Methods are well defined for a systematic review. It would be useful to clarify why men 15-49 were not included in the study, later in the Discussion it is explained that this data does not exist. But in the Methods line 119: "2. Sex: Male and female (PSAC and SAC); only female adults (WRA) [1]." it reads more like they were intentionally excluded from the review. Explaining earlier in the manuscript that this data does not exist would be useful for the reader.

Line 181 should be updated as there appears to be Six groups of categorization?

-Are the objectives of the study clearly articulated with a clear testable hypothesis stated? YES

-Is the study design appropriate to address the stated objectives? YES

-Is the population clearly described and appropriate for the hypothesis being tested? See comment above

-Is the sample size sufficient to ensure adequate power to address the hypothesis being tested? YES

-Were correct statistical analysis used to support conclusions? YES

-Are there concerns about ethical or regulatory requirements being met? NO, this is clearly explained

Reviewer #2: The methods are appropriate to the objetives

**Results**

-Does the analysis presented match the analysis plan?

-Are the results clearly and completely presented?

-Are the figures (Tables, Images) of sufficient quality for clarity?

Reviewer #1: -Does the analysis presented match the analysis plan? - YES

-Are the results clearly and completely presented? - YES

-Are the figures (Tables, Images) of sufficient quality for clarity? - Some improvements to the figures can be made, suggestions below.

Table 1 it looks as if the age 15 years is present both the SAC and WRA categories. Should WRA not start at 16 years? Especially since line 261 bases "median value for 15 yo for SAC" It would seem including this in both upper and lower bounds for two different categories causes a few more subjects being under- or over-dosed based?

Figure 3, is it possible to label the panels? The Y-axis states "ugr/kg" and "microg/kgr". Neither of these are correct and should be "ug/kg" this is applied inconsistently throughout the manuscript, please update. Do not abbreviate axis labels. Can the figures be saved as vector based, they appear very pixelated even for the text.

Figure 4, Y-axis needs to be clarified, Proportion of what? X-axis should be "Dosing Method" not "IVM dose regimen". Can the figures be saved as vector based, they appear very pixelated even for the text. Can the panels be labeled.

Figure 5, Panel A figure is really impactful and shows why the height and weight based dosing strategies are so misaligned, nice job! Is it possible to separate male and female subjects, perhaps in a separate figure? Panel B write out what 5, 50, 95 means. Panel C does not appear necessary to extend to 800 ug/kg, but there is a tiny pixel floating perhaps where a 5 or 6 year old subject would be ??

Can a similar breakout of age/dose by year be made for SAC?? Can the results for PSAC and SAC children be separated by sex? There are established differences in age/height/weight between boys and girls which could influence these dosing results. It would appear that your data set is large enough to assess this issue of sex differentiation.

Reviewer #2: There are some discrepancies with the PROSPERO registry. See below

**Conclusions**

-Are the conclusions supported by the data presented?

-Are the limitations of analysis clearly described?

-Do the authors discuss how these data can be helpful to advance our understanding of the topic under study?

-Is public health relevance addressed?

Reviewer #1: -Are the conclusions supported by the data presented? YES

-Are the limitations of analysis clearly described? YES

-Do the authors discuss how these data can be helpful to advance our understanding of the topic under study? YES

-Is public health relevance addressed? YES

Citation #18 does not appear correct, #19 link is broken. Find a better citation to support the claim of 500M people treated annually.

Reviewer #2: The conclusions are supported by data

**Editorial and Data Presentation Modifications?**

Reviewer #1: (No Response)

Reviewer #2: (No Response)

**Summary and General Comments**

Reviewer #1: This is a very important manuscript that advances a critical concept to streamline ivermectin mass drug administration. The paper is clear and presents very useful comparison of weight-, height-, age- based fixed dosing.

Can the results for PSAC and SAC children be separated by sex? There are established differences in age/height/weight between boys and girls which could influence these dosing results. It would appear that your data set is large enough to assess this issue of sex differentiation.

Reviewer #2: PNTD-D-25-00581

Fixed-Dose Ivermectin for Mass Drug Administration: Is it time to leave the dose pole behind? Insights from an Individual Participant Data Meta-Analysis. By Echazu et al.

This is a metanalysis conducted using individual patient data, evaluating the proportion of children in different age groups and women of reproductive age that receive the target dose of ivermectin based on anthropometric criteria. The authors propose fixed doses for the different target groups. A large dataset from countries endemic for STHs and filariae was used.

The manuscript is well written. The methods are appropriate for the objective and the conclusion is supported by the results.

Below I provide a few relevant comments

L75 “a clear upper safety limit for ivermectin in humans has not yet been formally established”

This is true but could be misleading to the non-expert reader. The studies of Kamgno some 20 years ago show good tolerability in adults but also a dose-dependent rate of AEs, particularly once the 800 μg/kg dose was exceeded. These AEs were not serious and indeed were mostly transient but could be the early signs of CNS toxicity. Buonfrate et al, saw a dose dependent AE rate when dosing either 600 or 1200 micrograms per kilo for five days. This is particularly worrisome for the neurologic category including Includes dizziness, headache, paraesthesia and somnolence.

Are these signs “safe”? that would be a good question for a regulatory agency, which should take into account the harm done by under-dosing in terms of target diseases and even potential resistance in the involved pathogens caused by sub-optimal systemic exposure. .

So, while the upper limit is not formally established, data seems to indicate the lower range is close to or at 800 μg/kg.

L197. Outcomes

There is a discrepancy with the PROSPERO registry which states: “Proportion of children receiving correct dose of ivermectin (200 to 600 µg/kg) with FDC compared with height or weight-based dosing”

The manuscript states 200-400

L199. “A secondary outcome was to compare the performance of the fixed-dose regimen with height or weight-based dosing regimens. The measure of effect for this outcome were proportion and median doses differences between dosing regimens”

This is part of the primary in Prospero, not a secondary.

What about the third outcome listed in PROSPERO? “Threshold age for contraindication of IVM”. What is the conclusion here?

L226 What do you mean by “Exclusions were performed randomly based on predefined criteria”. Were data excluded at random?

L279-281: “The proportion of participants classified as receiving doses above the recommended range was higher with the fixed-dose regimen (11.27%), a difference that was statistically significant compared to the other regimens”

These data (% receiving higher than recommended doses) should be presented

L281. 0.02% of 500 million treated yearly is 100.000 exceeding 800 mcg/kg yearly. This should not be downplayed.

L301. A version of figure 3 panel B but by category (PSAC; SAC and WRA) would be informative

L321-323 I don’t quite see the “lack of concordance” as the pole limit of 90 cm effectively prevents approximately 95% of children under 15kg to receive treatment, hence fulfilling the current indication. It would be interesting however to quantify the proportion of children under 90cm who weight more than 15kg and are wrongly excluded from treatment using the pole.

Discussion:

In general, is balanced and appropriate. It would be informative to include a paragraph on risk-management in those receiving higher doses.

Additional potentially important points:

What is the suggestion for children of uncertain age?

The authors report stunting rates exceeding 48% in some target countries. It would be valuable to mention/discuss the importance of protein binding and distribution of ivermectin in severely malnourished children. Is there any data regarding blood-brain barrier integrity and malnutrition?

Mentioning the case report from France of a child with a nonsensical mutation in the Pgp leading to an ICU admission could be useful as a reference to a rare event (PMID: 32813957)

Data accessibility:

I could not access the data at DOI: 10.6084/m9.figshare.28734971.

PLOS authors have the option to publish the peer review history of their article (what does this mean?). If published, this will include your full peer review and any attached files.

Reviewer #1: No

Reviewer #2: No

**Figure resubmission:**
---

## [Decision Letter · Decision Letter 1]

27 Aug 2025

Dear Dr Krolewiecki,

We are pleased to inform you that your manuscript 'Fixed-Dose Ivermectin for Mass Drug Administration: Is it time to leave the dose pole behind? Insights from an Individual Participant Data Meta-Analysis.' has been provisionally accepted for publication in PLOS Neglected Tropical Diseases.

Best regards,

Michael Marks

Academic Editor

Francesca Tamarozzi

Section Editor

Shaden Kamhawi

co-Editor-in-Chief

Paul Brindley

co-Editor-in-Chief

Reviewer's Responses to Questions

**Key Review Criteria Required for Acceptance?**

**Methods**

-Are the objectives of the study clearly articulated with a clear testable hypothesis stated?

-Is the study design appropriate to address the stated objectives?

-Is the population clearly described and appropriate for the hypothesis being tested?

-Is the sample size sufficient to ensure adequate power to address the hypothesis being tested?

-Were correct statistical analysis used to support conclusions?

-Are there concerns about ethical or regulatory requirements being met?

Reviewer #1: -Are the objectives of the study clearly articulated with a clear testable hypothesis stated? - YES

-Is the study design appropriate to address the stated objectives? - YES

-Is the population clearly described and appropriate for the hypothesis being tested? - YES

-Is the sample size sufficient to ensure adequate power to address the hypothesis being tested? - YES

-Were correct statistical analysis used to support conclusions? - YES

-Are there concerns about ethical or regulatory requirements being met? - YES

Reviewer #2: All my comments have been appropriately addressed

**Results**

-Does the analysis presented match the analysis plan?

-Are the results clearly and completely presented?

-Are the figures (Tables, Images) of sufficient quality for clarity?

Reviewer #1: -Does the analysis presented match the analysis plan? - YES

-Are the results clearly and completely presented? - YES

-Are the figures (Tables, Images) of sufficient quality for clarity? - YES

Reviewer #2: All my comments have been appropriately addressed

**Conclusions**

-Are the conclusions supported by the data presented?

-Are the limitations of analysis clearly described?

-Do the authors discuss how these data can be helpful to advance our understanding of the topic under study?

-Is public health relevance addressed?

Reviewer #1: -Are the conclusions supported by the data presented? - YES

-Are the limitations of analysis clearly described? - YES

-Do the authors discuss how these data can be helpful to advance our understanding of the topic under study? - YES

-Is public health relevance addressed? - YES

Reviewer #2: All my comments have been appropriately addressed

**Editorial and Data Presentation Modifications?**

Reviewer #1: No

Reviewer #2: All my comments have been appropriately addressed

**Summary and General Comments**

Reviewer #1: (No Response)

Reviewer #2: All my comments have been appropriately addressed

PLOS authors have the option to publish the peer review history of their article (what does this mean?). If published, this will include your full peer review and any attached files.

Reviewer #1: No

Reviewer #2: No

---

## [Editor Report · Acceptance letter]

Dear Dr Krolewiecki,

We are delighted to inform you that your manuscript, "Fixed-Dose Ivermectin for Mass Drug Administration: Is it time to leave the dose pole behind? Insights from an Individual Participant Data Meta-Analysis.," has been formally accepted for publication in PLOS Neglected Tropical Diseases.

Best regards,

Shaden Kamhawi

co-Editor-in-Chief

Paul Brindley

co-Editor-in-Chief
